**Article** https://doi.org/10.1038/s41467-022-32724-z

# Super-resolution laser probing of integrated circuits using algorithmic methods

V. K. Ravikumar[1,2], Jiann Min Chin[2], Winson Lua[2], Nathan Linarto[2], Gopinath Ranganathan[2], Jonathan Trisno [1,3], K. L. Pey [1] ✉ & Joel K. W. Yang [1,4] ✉

Laser probing remains invaluable to the semiconductor industry for isolating and diagnosing defects in silicon transistors in integrated circuits during electrical stress tests. However, continuous device miniaturization below the 20 nm technology node has crammed multiple transistors within the focal spot of the laser beam, resulting in signal crosstalk, poor beam positioning accuracy and degraded fault isolation capabilities. The challenge is analogous to focusing attention to a single speaker in a crowd despite the multiple simultaneous conversations in the background. Through algorithms introduced in this patented work, consisting of cross-correlations, clustering, and our previously developed combinational logic analysis, we achieved beam positioning accuracy to better than 10 nm, extracted electrooptic waveforms from a node of a group of transistors (~18 times beyond the optical resolution limit), and applied this to isolate and identify an actual fault on a defective device. While problems associated with probing with shorter wavelength lasers continue to be addressed, our approach enhances and enables the continued probing of ICs using sub-bandgap photon energies without hardware modification to existing technology at semiconductor technology nodes below 10 nm.

Modern integrated circuits have achieved unprecedented performance for a given cost or power consumption over previous generations due to constant and deliberate improvements in the manufacturing processes gained from learning from and understanding failure mechanisms. Laser probing (LP) is a fault localization technique that uses a near infrared optical beam that is tightly focused by a solid immersion lens system to measure minute changes in the optical absorptance of transistors. Transistors are probed while the diagnostic electrical tests are run on the processor, and aid in identifying a failing transistor or interconnections (net) non-destructively[1–8]. A logical approach to keep up with the ever-shrinking dimensions of transistors is to use shorter wavelengths in laser probing. However, the strong absorption of light by Si substrate for photon energies greater than Si bandgap limits the resolution of the solid immersion system[9–11] by Abbe's criteria to ~180 nm with 1064 nm lasers[12]. At the 7-nm technology node where the transistor size of ~55–60 nm, with a drawn gate width <10 nm[13], this resolution limit constraints the ability to optically locate and accurately align the optic probe with the transistor of interest. In addition, multiple neighboring signal sources could interact with the optic probe simultaneously, and the recorded waveforms are affected by this crosstalk, making interpretation a challenge. Due to the limited optical resolution for probe alignment and electrooptical crosstalk, locating and probing on individual transistors within the device have become increasingly difficult, resulting in arduous

[1]Singapore University of Technology and Design, 8 Somapah Road, Singapore 487372, Singapore. [2]Advanced Micro Devices (Singapore) Pte Ltd, 508 Chai Chee Lane, Singapore 469032, Singapore. [3]Institute of High Performance Computing, Agency for Science, Technology and Research, 1 Fusionopolis Way, Singapore 138632, Singapore. [4]Institute of Materials Research and Engineering, Agency for Science, Technology and Research, 2 Fusionopolis Way, Singapore 138634, Singapore. ✉e-mail: peykinleong@sutd.edu.sg; joel_yang@sutd.edu.sg

searches for the defect, reducing success rates of Failure Analysis (FA) of integrated circuits at these technology nodes. Thus, while LP with shorter wavelengths (such as visible light)[14, 15] enhances optical resolution by up to two-folds and reduce crosstalk, they are affected by large Si absorption restricting Si substrate thicknesses and generate free carriers in Si which reduces the quality of the collected signals[16].

Previously developed combinational logic analysis (CLA) algorithms[17–20] help to predict the shape of the ensembled signals. Although CLA was useful in simple combinational cells in mature technology, the methods described were neither scalable to smaller technology nodes, nor automatable, posing fundamental limitations to widespread adaptation. In this work[19, 21], we introduce an algorithmic method consisting of CLA, cross-correlations[22], and clustering[23] named CCC (one C for each of the three aforementioned algorithms) that leverages on the known available electrical information to aid in aligning the optic probe and extracting the signals of interest. Analogous to the selective auditory attention effect[24], where prior knowledge of the relative position, the pitch, and script of each speaker would allow one to tune into a particular "speech", we leverage on prior knowledge of electrical and optical information to accurately align the optic probe with the signal source of interest, and extract signals from densely packed elements at the single-transistor and sub-transistor resolution, up to 18-times smaller than the diffraction limited probe spot. The CCC allows scalability and can be combined with scripts to control the laser probe to automate, and extract volumes of useful information from the target circuitry. The results of this work provide a different perspective to imaging optoelectronic systems with compromised spatial resolution and extends the utility of infrared laser probing to sub-10 nm technology nodes.

We performed LP on fin-shaped three-dimensional transistors known as FinFETs[25] using a solid immersion laser scanning microscope (LSM) by electrically stimulating through the interconnects to the transistors (top) while the optic probe is focused through the Si substrate (bottom) for collection of optical signals, as shown in Fig. 1a. Figure 1b is an optical micrograph of a 7-nm-node device collected through the LSM. As observed, even relatively large alignment features such as boundaries (isolation) between transistor arrays cannot be discerned. Despite the mechanical stability of the laser scanner and accuracy in probe deflections (sub-10 nm at highest LSM zoom and SIL objective), the poor optical resolution leads to ambiguity in determining the physical position of the probe relative to the transistor or the node of interest when compared with a computer aided design (CAD) of the circuit layout. The red box marks an area of a cell to be probed, consisting of over 40 transistors within a 670 nm × 480 nm area. A cell, which is a fundamental building block of integrated circuits is made up of a group of transistors (up to few dozens) and performs a specific logic operation on the inputs. Nets are nodes of transistors linked by interconnects and share the same electrical potential. Figure 1c is a scanning electron micrograph (SEM) of the cell with its CAD layout overlaid. The red circle indicates the size of the diffraction limited laser probe spanning multiple transistors and nets within the cell. Optical measurements track minute changes to the intensity of light reflected from the transistors that correlate to the absorption by free carriers that vary as transistors are switched on and off, as given by Eq. 1[26].

$$\triangle\alpha = \frac{\lambda^2 q^3}{4\pi^2 C_0^3 \varepsilon_0 n_u}\left(\frac{\Delta N_e}{m_e^2 \mu_e} + \frac{\Delta N_h}{m_h^2 \mu_h}\right) \tag{1}$$

where $\Delta\alpha$ is the change in the absorption coefficient, $\lambda$ is the wavelength of light, $q$ is the charge of an electron, $C_O$ is the speed of light in vacuum, $\varepsilon_O$ is the permittivity in free space, $n_u$ is the refractive index of undoped silicon, $\Delta N_e$ and $\Delta N_h$ are changes to carrier densities, $m_e$ and $m_h$ are effective masses, and $\mu_e$ and $\mu_h$ are the mobilities of electrons, and holes respectively. As described by Soref and Bennet, for a large change $\Delta N_e$ of $10^{17}$ cm³, $\Delta\alpha$ changes by 0.1 cm⁻¹. As the interaction

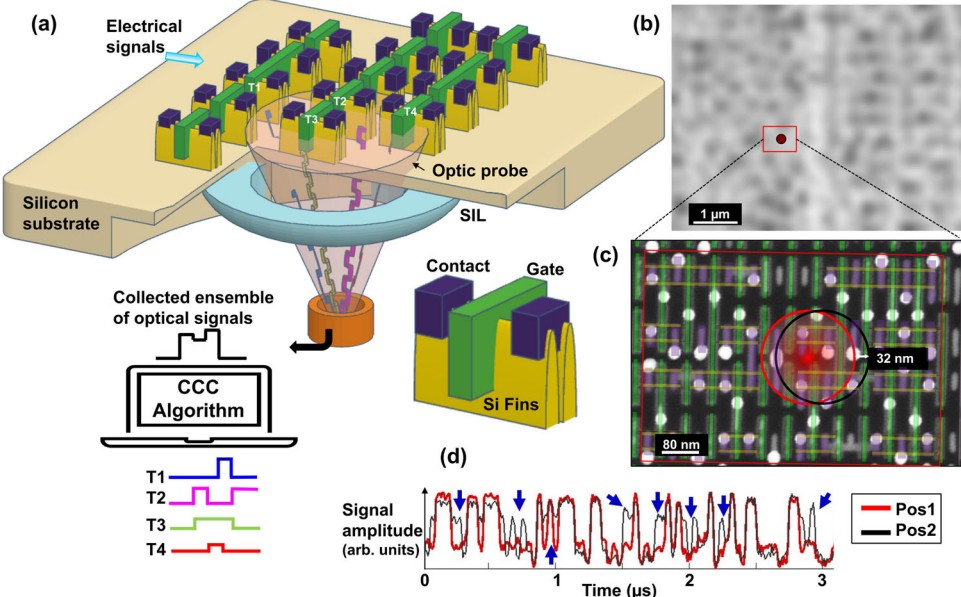

**Fig. 1 | Laser Probing with algorithmic analysis to determine accurate probe position. a** Schematic of LP signal collection on an IC built on sub-10 nm FinFET technology. An optic probe is focused onto the active circuit to measure changes in the reflectance while electrical tests are executed. As multiple transistors interact with the optic probe simultaneously, the collected ensemble of optical signals from transistors T1-T4 is processed by our CCC algorithm to extract signals from individual transistors. **b** High-resolution LSM reflected micrograph using 2.9 NA and 1064 nm wavelength on the area of interest is unable to resolve features useful for CAD alignment due to the limited optical resolution. The red box indicates the cell area to be probed and the circle within indicates the size of the optic probe spot. **c** SEM micrograph of the cell area to be probed overlaid with the CAD layout of the FinFET, gates (green), fin (yellow), and contact (violet). The bright circles in the micrograph are vias which route electrical signals to transistors. The red and black circles indicate the approximate areas of interaction of the optic probe with the circuit. **d** LP signal waveforms collected at ~32 nm from each other (Pos1 and Pos2) in a 3-μs time window. Multiple levels exist in each waveform because of the ensemble of individual signal sources under the optical probe. The blue arrows indicate instances in time where the waveforms differ from each other.

volume of light with the affected free carriers is microscopic, the change in reflected power is therefore only a few microwatts, which necessitates the averaging over millions of test loops to improve the signal to noise ratio. A detailed description of the flip chip technology[27], setup for dynamic fault isolation[28], and signal generation in laser probing systems[29] is available in literature.

LP waveforms are created by recording changes to the amplitude of reflected light, triggered by carrier fluctuations within the active regions of the transistor as the transistor switches between electrical states. Figure 1d shows laser probed waveform traces from two positions (red and black) that are collected from physical locations merely 32 nm apart, and their corresponding approximate positions are marked in Fig. 1c. The waveforms illustrate the aforementioned challenges: (1) The waveforms exhibit multiple intensity levels as marked by blue arrows. While the electrical state of a digital circuitry is binary, i.e., either ON or OFF, these intermediate intensity levels occur due to the superposition of signals from multiple transistors (or electrooptical crosstalk) within the optic probe, and their respective modulation capacity (MC) as discussed later. Decomposition of the ensemble signal into its individual transistor or net constituents, ie, overcoming the electrooptical crosstalk, is important for waveform interpretation and analysis. (2) A displacement in the position of the optical probe by 32 nm (<18% of the optical resolution) already shows significant deviations in the waveform, underscoring the importance of accurate probe positioning despite the lack of optical resolution. The method we propose will solve these challenges by extracting the signals from the different individual transistors through the proposed CCC algorithm.

## Results

### Positional cross-correlations for accurate probe placement

To improve the accuracy of alignment between the physical device with CAD, we introduce a method to achieve sub-10 nm level of beam positioning accuracy through a process of correlation between simulations and actual waveforms collected from the cell. As a cell is made of many transistors, a positional cross-correlation helps to ensure that the waveform collected is a representation of the transistors within the cell that are of interest. A multitude of LP waveforms (raw) is collected from different positions ($x_r$) within the cell and is correlated using Pearson's correlation coefficient (PCC) to CLA simulation waveforms on different positions ($x_s$) within the cell, ie, the position of the CLA simulation waveforms is treated as points of reference to pinpoint the location of the raw waveform. PCC is calculated as described in Eq. 2[22].

$$PCC = \frac{cov(CLA, Raw)}{std(CLA)*std(Raw)} \quad (2)$$

A complex cell comprising of about 24 transistors built with sub-10 nm technology with a MC is shown in Fig. 2a. The optic probe spans ~190 nm and interacts with multiple transistors at any physical position. A multitude (23 counts) of raw waveforms are collected from the physical device by positioning the probe at ~32 nm step size from left to right as indicated by the 'X's. Each of these waveforms are correlated one at a time with CLA simulations performed along the same axis at 1 nm increments. Simulations along the longer axis for the cell of interest can be completed within a minute on a modern computing workstation, which can be accelerated proportionally by increasing the step size.

The correlation method allows the user to pin-point the exact location of the transistor of interest with an accuracy of 10 nm. As the device is made with 7 nm technology, sub-transistor structures such as the gates and the drains are dimensionally much smaller than the optic probe as shown, contributing to the probe placement resolution. As an example, two neighboring raw waveforms r1 and r2 (locations indicated by purple and green dots respectively) are shown in Fig. 2(b).

The raw waveforms are compared to the CLA simulation waveforms of different positions. The best correlated CLA simulations to r1 and r2 are traced in black. The red and blue dashed traces are simulations 10–30 nm away from the best correlated simulation. A strong deviation in relative amplitudes can be observed in these simulations despite a displacement of just 5–15% of the optic probe size. This strong deviation forms the basis of the probe positioning accuracy and is derived from the changes offered by the fluctuations in MC.

The placement resolution is realized through thresholding the PCC distribution. As shown in the heat map (Fig. 2c), a range of simulation positions can strongly correlate with each raw waveform. The position dependence of the spread in PCC profile is shown in Fig. 2d, where the PCC spread (pcc1 – pcc4) for four waveforms are overlaid by aligning the peak PCCs together. The position at which these waveforms were collected are marked in Fig. 2b. The PCC spread profiles appear to be similar or worse than the distribution of the optic probe. A placement resolution better than 10 nm can be achieved on pcc3 and pcc4 when a thresholding of 0.1% is applied (Supplementary Table 1). The super-resolution positioning accuracy is achieved through the accurate determination of the location of the peaks of a diffraction limited point spread function, reminiscent of fluorescence microscopy techniques such as stochastic optical reconstruction microscopy[30] though we have no fluorescence in this context.

However, some positions may not be resolved with such accuracy, as resolution depends on the spatial variation in MC and is reduced if neighboring circuitry do not offer variations in MC, such as parallel transistors. The broader distributions of pcc1 and pcc2 are likely due to unavailability of distinct electrical signals from the local area and could only be resolved to 34 nm and 16 nm, respectively. Even at 34 nm, the placement resolution has improved to more than five times the optical resolution limit of the tool.

### Signal extraction of a single net

Once the CAD is aligned with the help of the methods described in the previous section, every circuitry in the field of view is automatically also aligned to CAD, making navigation to any circuitry within the view easy and accurate. In addition to accurate probe placements, CLA simulations are useful in extracting signals from individual nets within a probed area and in understanding if the observed waveforms are defective. In the following section, we discuss the signal extraction methodology, and the application of simulations to a failure analysis performed on a defective microprocessor at sub-10 nm technology.

To extract a signal from a single net ($R_{net}$), a raw waveform ($W_o$) is first collected from the area of interest, preferably centered at, or containing the net and is processed with signal subtraction techniques as described below. Through positional correlations described in the previous section, the accurate physical location is determined. Then, the crosstalk signal ($S_{CT}$) is simulated by removing the electrical information to the net of interest and calculating the signals from the environment of the net. When the crosstalk waveform is scaled appropriately and subtracted from the raw waveform, it can produce the signal of interest. As the simulation and raw waveforms are normalized, a direct subtraction between the $W_o$ and $S_{CT}$ may not yield $R_{net}$ directly. A time-invariant scaling parameter ($C_{net}$) is introduced to extract $R_{net}$ from $W_o$ and $S_{CT}$ as shown in Eq. 3.

$$R_{net}(t) = W_o(t) - C_{net} \cdot S_{CT}(t) \quad (3)$$

As the nets of interest in our circuitry is expected to have only two levels under normal operation, the $C_{net}$ can be determined through a combination of clustering algorithms[23] that separate the waveform into two clusters while minimizing a loss function ($Loss_1$) shown in Eq. 4. Each cluster is formed by applying a k-means clustering algorithm[23] to $R_{net}$, which is commonly used to split a distribution into k number of clusters. The algorithm clusters the waveform into two

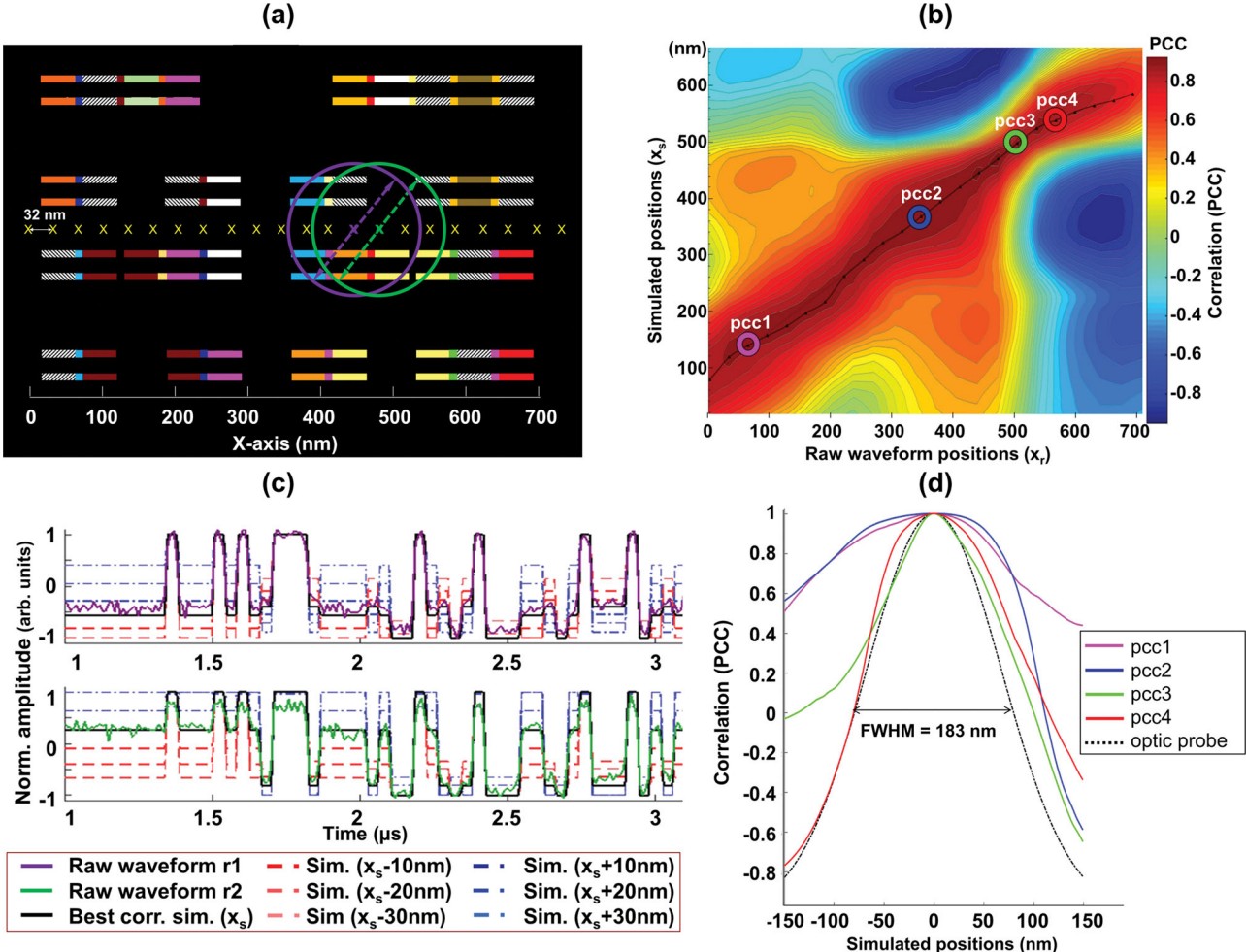

**Fig. 2 | Accurate probe positioning at the target transistor. a** Physical layout of cell (670 nm wide) under observation. The tiny squares represent the gate-fin overlap and the longer horizontal bars represent the source or drain segments of the fin. The polygons with the same color fluctuate with the same electrical signal while polygons with diagonal textures are power supplies and do not toggle. The yellow 'X's represent the raw waveform positions ($x_r$) from which each of 23 optically probed (raw) waveforms are collected within the cell. Raw waveforms r1 and r2 are collected from positions indicated by the violet and green circles with diameters equal to the FWHM of the probe. **b** LP waveforms traces comparing r1 and r2 with CLA simulations. The best correlated simulations positions ($x_s$) for r1 ($x_s$ =471 nm) and r2 ($x_s$ = 497 nm) are colored in black. Overlaid are also simulations at positions $x_s$ −10 nm, −20 nm, and −30 nm in red and $x_s$ +10 nm, +20 nm,

and +30 nm in blue. **c** Heat map of the positional correlation between the raw waveforms ($x_r$) on X-axis (32 nm increments) with simulations ($x_s$) performed along the same axis with a resolution of 1 nm on Y-axis. A strong positive correlation is represented in deep red, and a strong inverse correlation is in deep blue. As the positions of the optic probe is changed along the X-axis, the physical positions correlate well with the corresponding positions in the simulation. The trend line connects the best correlated $x_s$ with $x_r$. **d** Graph compares the variation in PCC index with the simulation positions for four physical positions (pcc1–pcc4) marked by colored rings in **c**. In dotted black is the Gaussian optic probe distribution, which is sharper than the PCC distributions of each of the correlations, indicating that the method does not violate the optical diffraction limit of the system.

groups (k = 2). Clusters cl(H) and cl(L) represent the ON and the OFF states of the net.

$$Loss_1 = std(cl(H)) + std(cl(L)) \qquad (4)$$

Optimizing recursively between Eqs. 3 and 4 using gradient descent[31], this artificial intelligence (AI) algorithm can identify the ideal $C_{net}$ for the smallest $Loss_1$. Alternate methods such as linear search with coarse and fine step sizes or even trial and error may also be used to identify a suitable $C_{net}$. Figure 3a shows a portion of the layout of the same cell that was previously described. Raw waveforms (r1–r4) are collected from positions marked brown, green, purple and magenta, respectively. These waveforms are position correlated using methods described in the previous section. Three nets n9 (blue), n15 (orange), and n16 (red) under the optical probe are investigated. Net n9 comprises 4 fins, each 50 nm long and <10 nm wide, n15 comprises 2 fins, each 50 nm long and <10 nm wide, and n16, the smallest net, comprises

2 gates each with a drawn width and length of about 10 nm. These signals are extracted by calculating $R_{net}$ via Eqs. 6 and 7. The extracted waveforms for each net n9, n15 and n16 from the raw waveform r2 are shown as the colored waveforms in Fig. 3b.

The extracted waveforms for each net n9, n15, and n16 are shown in Fig. 3b, and their correlations as a function of position tabulated in Fig. 3c. The following observations can be made: First, the extracted waveforms for net n9 and n15 correlate well with the expected timing diagram (black traces) in Fig. 3b. As shown in the correlation values as a function of position in Fig. 3c, a strong correlation more than 90% is observed for the larger nets, n9 and n15. The correlation of n9 (97%) is larger than n15 (90%) likely because n9 is larger than n15. Likewise, the weakest correlation is observed for smallest net, n16. Second, although weak, a correlation of 58.9% on net n16 is still significant. Remarkably, by extracting the signal from a net with a drawn width and length of about 10 nm, we show that the method can isolate a net 18 times smaller than the optical probe's diameter, or about 125 times smaller

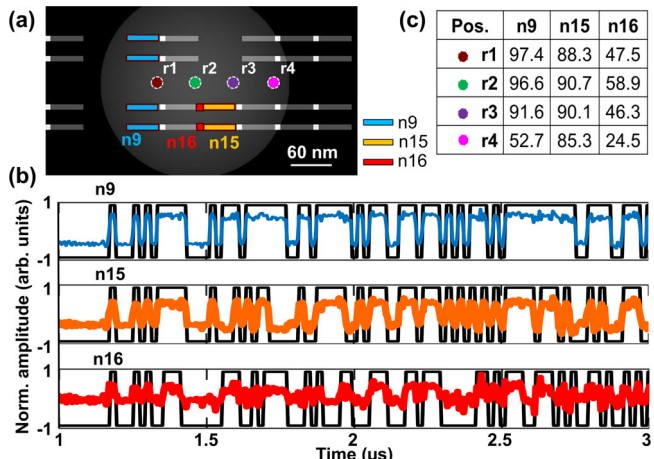

| Pos. | n9 | n15 | n16 |
|------|------|------|------|
| ● r1 | 97.4 | 88.3 | 47.5 |
| ● r2 | 96.6 | 90.7 | 58.9 |
| ● r3 | 91.6 | 90.1 | 46.3 |
| ● r4 | 52.7 | 85.3 | 24.5 |

**Fig. 3 | Signal extraction down to the gate level. a** A section of the layout under observation. Four raw waveforms are collected from positions marked by circles (*r1–r4*). Nets *n9, n15,* and *n16* are collocated within the optic probe in each of the positions. **b** Graph showing extracted waveforms from *n9, n15,* and *n16,* from *r2* overlaid with expected timing diagrams in black. **c** Table summary of the highest correlations (pcc) of each extracted net with its corresponding timing diagram shows that the strongest signal-to-noise ratio of the extracted signals is achieved on the largest physical net *n9,* and occurs when the optic probe is positioned most accurately on the net (at *r1*). Remarkably, the signals from a gate net *n16* which spans <200 nm² in area can be extracted with a 58.9% correlation to the expected signal at *r2* despite the optic probe spanning a lateral area >25,000 nm².

area. As the net size reduces, the extracted signal becomes noisier. The noise in the waveform extracted from *n16* is high because the signal intensity from the tiny net is weak in comparison to the electronic and optic noise of the system. Finally, the best correlated position for *n9* is at *r1*, for *n15* is *r2/r3*, and *n16* is at *r2*. When compared to their physical positions in Fig. 3a, it is apparent that the positional placement contributes to the variation in signal strength of the extracted signal. The signals from individual nets can be extracted best when the optical probe is most accurately cantered on the net. These results agree well with expectations, as the optical probe is expected to be modulated most by the element when the peak power of the probe interacts most with that element.

## Defect prediction

CLA simulations most importantly are used to predict the defect location by comparing the collected raw waveforms with artificially injected failures within the cell. In this section, we detail a failure analysis case study performed at 7 nm technology node, taking advantage of these waveform simulations. The failing microprocessor was a reject from a manufacturing process and automated test equipment-based diagnostics revealed that a 3-input combinational logic cell from Fig. 2 could contain a defect. The inputs of the cell are numbered A1–A3 and the physical layout of this cell is shown in Fig. 4a. The laser was positioned using cross-correlations with a known good cell in the field of view. Raw waveforms were collected from approximately the center of the failing cell, and with a reference cell that was injected with the same series of inputs. The waveforms were compared with the simulations as shown in Fig. 4b. The bad waveform showed mismatches with the CLA and the good waveforms at multiple cycles, confirming the diagnostic results.

The waveform was segmented into constituent input combinations (eight segments for 3-input cells) and compared with the CLA amplitudes for each input. It was observed that when the input A2 was forced high, irrespective of other inputs were the most mismatched, and suggested that the defect could be located on a net associated with the input A2. As multiple defects could explain this observation,

over twenty potential defects of resistive shorts nature were selected for further analysis. Four such defects are shown in Fig. 4a, (marked in red and orange polygons, D1–D4). SPICE models[32] were generated to model the electrical behavior of each transistor/net within the cell containing each of these defects sequentially, and corresponding CLA simulations were conducted. It is worthwhile to note that while resistive shorts were investigated for this study, other defects such as open interconnects could also be considered.

The simulations with defects were correlated with the bad waveform (black trace) as shown in Fig. 4(c). The simulation with a defect D2, consisting of a resistive short between net A2 and VDD had the strongest correlation to the bad waveform. Some of the related correlations are summarized in Fig. 4d. Although the bad waveform correlated with the original CLA simulation with a strength of only 84%, it correlated with a simulation that included defect D2 overwhelmingly, with a strength of 96%. Though defects at other positions offered varying strength of correlations, the strongest, from our defect simulations were with D2 and was therefore the strongest candidate for physical analysis and material characterization.

Physical analysis was then performed by removing metal interconnects recursively with careful inspections of target circuitry. A 2-wire resistance measurement was performed using in-situ Nanoprober within a SEM at the contact layer as shown in Fig. 4e. The probes were landed on the gate and the drain of the transistor indicated by defect D2, that would normally exhibit a high (giga ohms) resistance and non-linear change in current conducted across the terminals. However, as shown in Fig. 4f, a low-resistance short was observed, which confirmed the root cause and supported the defect location hypothesis.

## Discussion

Continuing technology scaling requires a thorough feedback from failing devices that can help improve fabrication processes, reliability, and silicon design. Scaling beyond 20 nm technology node has become particularly difficult due to imaging resolution limits and increased complexity in failure analysis. In this work, we have shown conclusively that algorithmic methods and deduction can overcome some fundamental challenges posed by optical resolution limitations to laser probing on cutting edge technology electronic devices. Interestingly, the resolution is achieved through unique electrical signature within the target circuitry without violating the optical diffraction limit of the laser probing setup. As technology scales, more transistors and more variation in modulation capacity can be expected within the optic probe. Counterintuitively, the resolution of our technique can be expected to improve further with technology scaling due to the increased complexity of signals within the optic probe, limited only by fine displacement resolution of the laser scanning hardware. Hence, adopting these methods will be beneficial for failure analysis where the spatial resolution of the optical system is unable to keep up with the spatial resolution of lithography in future technology nodes.

As technology is scaled, the simulation load increases proportionally with the number of transistors in consideration. As the runtime for the simulation of 7 nm cells take only ~1–2 s per waveform, the process is extremely scalable with technology. Iterative loops of probe placement deflections and cross-correlations can be automated to be used for large volume signal extractions effectively. However, to successfully apply these methods to a large area, the precursor files such as the physical layout, SPICE models and test pattern stimuli need to be prepared in advance and is an area of active development.

## Methods
### Computation of modulation capacity
The amplitude of the LP waveform is a function of the variation in the reflected optic power. The total reflected power can be separated into

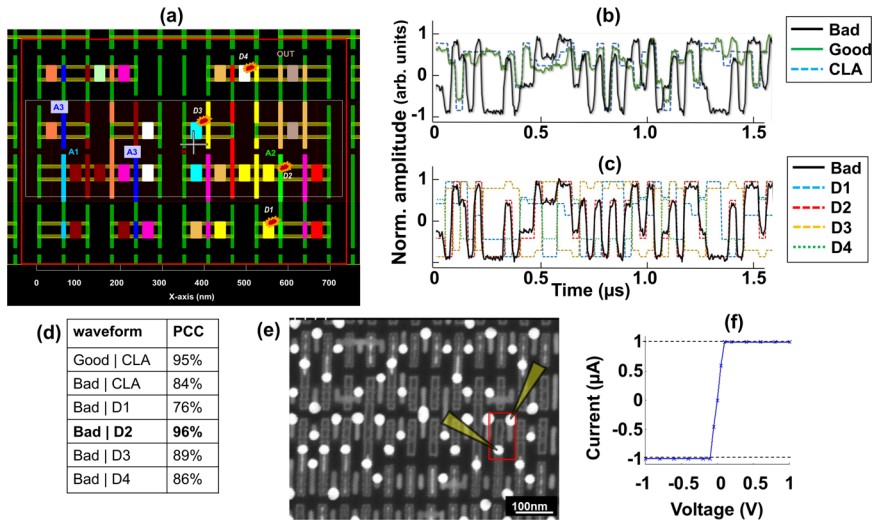

**Fig. 4 | Accurate defect location prediction using waveform simulation.**
**a** Physical layout of the failing cell showing internal connections. A1–A3 are inputs to the cell. Also highlighted are few potential defects (short)(D1–D4) that could explain the failure. **b** Raw waveforms collected from the failing cell (solid black) and an identical good cell (solid green), with the reference CLA waveform (dashed blue). The bad waveform has many mismatches, particularly when input A2 = 1. **c** Waveforms overlaid with CLA simulations of defective cells with defects D1– D4.

CLA waveform simulated with defect D2 (red dashed) matches the raw waveform (solid black) with >96% correlation. **d** Table shows strong correlation between bad raw waveform and CLA simulation with defect D2. **e** SEM micrograph of the suspected transistor probed electrically by 2-wire nanoprobe (probes positions indicated by yellow). **f** Current−Voltage curve confirms a resistive short between the suspect net A2 (gate) with VDD (source). The dashed lines represent compliance limits set at 1 μA.

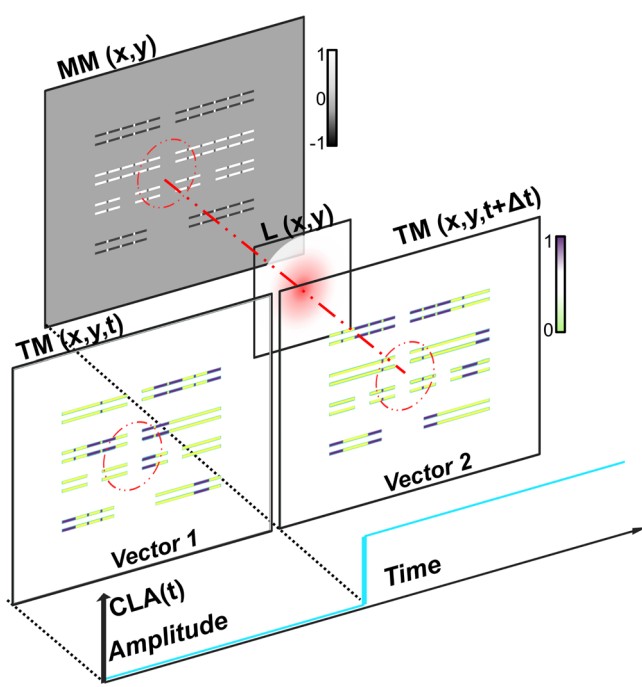

**Fig. 5 | Schematic of the MC and CLA simulation.** MM contains the relative modulation that the device imparts to the optic probe L. TM changes with each input stimulus, affecting the modulation over time. MC combines MM and TM to contain a time variant map of the relative modulations. A simple integration of the dot product between MC and L produces the dynamic reflected power, which is presented as the simulated waveform CLA(t).

static (does not change) and dynamic components (relative changes with electrical stimuli). By estimating the changes to the dynamic components, we can predict the shape of the ensemble probed waveforms. The process can be explained with the help of the schematic shown in Fig. 5.

First, we construct a high-resolution (1 nm in $x$ and $y$) two-dimensional model of the circuitry's physical layout called the modulation map (MM) which represents the spatial distribution of the relative modulation that the device would impart to the optic probe. The map includes all active areas where relative changes to free carriers are expected, such as the gate, source and drain regions of each transistor, and is normalized to the PMOS drain as a convention. Second, we construct a timing map (TM) of the same circuitry, which combines the electrical models of the circuitry and test stimuli sequences. Since the test vectors change with time, the timing map can be construed as a three-dimensional matrix, with the third dimension representing time. A piecewise multiplication of the modulation map with the timing map is a time-varying three-dimensional matrix called the modulation capacity (MC)[33] and is described in Eq. 5. The MC thus represents the influence of the physical device on the optic probe over time.

Next, we model the optic probe as a Gaussian distribution. The full width at half-maximum spread (FWHM) of the probe is related to the wavelength ($\lambda$) and numerical aperture (NA) using Abbe's criteria, as shown in Eq. 6[9]. Moreover, the standard deviation ($\sigma$), is related to the FWHM described in the same equation[34]. Therefore, $\sigma$ can be represented in terms of $\lambda$ and $NA$ as shown in Eq. 7. For the system used in this work, $\lambda = 1064$ nm, $NA = 2.9$, FWHM is 183 nm, and $\sigma = 77$ nm. The optical intensity distribution ($L$) centered at position ($x_0$, $y_0$) is shown in Eq. 8. The reflected power from a particular pixel ($x$, $y$) at a given time is a function of the product of the modulation capacity of the pixel at that time with the optical power on that pixel. The total dynamic reflected power is calculated by integrating the reflected power from every pixel of interaction between the device and optic probe as shown in Eq. 9. The ensemble waveform, which we call CLA waveform, is proportional to the dynamic reflected power. It is normalized between −1 and 1, and can then be plotted as a function of time.

$$MC(x, y, t) = MM(x, y) \cdot TM(x, y, t) \qquad (5)$$

$$FWHM = 0.5\frac{\lambda}{NA} = \sigma.2\sqrt{2\ln2} \qquad (6)$$

$$\sigma = 0.21 \frac{\lambda}{\text{NA}} \tag{7}$$

$$L = \frac{1}{\sqrt{2\pi\sigma^2}} e^{\frac{-((x-x_0)^2 + (y-y_0)^2)}{2\sigma^2}} \tag{8}$$

$$\text{Dynamic reflected power}(x_0, y_0, t) \propto \iint_{x_i, y_j = 0}^{n} \text{MC}(x_i, y_j, t) \cdot \text{L}(x_i, y_j) \, dx. \, dy \tag{9}$$

## Data availability
Due to the nature of this research, participants of this study did not agree for their data to be shared publicly, so supporting data is not available.

## Code availability
Due to the nature of this research, participants of this study did not agree for their code to be shared publicly.

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

## Acknowledgements

The work is funded by Singapore Economic Development Board and Advanced Micro Devices Inc. through the IPP program [P.K.L.]. © 2022 Advanced Micro Devices, Inc. All rights reserved. AMD, the AMD Arrow logo, and combinations thereof are trademarks of Advanced Micro Devices, Inc. Other product names used in this publication are for identification purposes only and may be trademarks of their respective companies. This research was also supported in part by National Research Foundation (NRF) Singapore, under its Competitive Research Programme award NRF-CRP20-2017-0004 [J.K.W.Y.] and NRF Investigatorship Award NRF-NRFI06-2020-0005 [J.K.W.Y.].

## Author contributions

V.K.R., P.K.L., and J.K.W.Y. conceived the idea. V.K.R., W.L., N.L., G.R., and J.T. performed the tests, calculations, and analyzed the data. V.K.R. and J.M.C. contributed the test facilities and collected the data. V.K.R., J.T., P.K.L., and J.K.W.Y. drafted and finalized the manuscript. All the authors discussed the results and commented on the manuscript.

## Competing interests

The authors declare no competing interests
