## [Peer Review File · Nature Communications]

REVIEWER COMMENTS

Reviewer #1 (Remarks to the Author):

In this work, a new technique is presented to address the issue of decreasing tech node size alongside a relatively stagnant laser spot size/resolution in laser probing. Laser probing is important for locating and observing fabrication faults in ICs, so the technique has the potential to be impactful in the field of failure analysis. The technique, referred to by the authors as the “CCC algorithm”, consists of a three-part process in the form of Cross correlations, Clustering, and Combinational logic analysis. First, a combinational logic analysis waveform is generated, by constructing a spatial activity map over time from the target circuitry’s layout and incorporating equations pertaining to optical probing intensity. Then, the positional accuracy of the probing laser is finely-tuned to ensure that the user’s collected waveform genuinely corresponds to the transistors of interest. This is accomplished by using Pearson’s correlation coefficient to correlate collected waveforms at different positions with the expected simulation waveform. Third, a method is performed to extract a signal from a particular net within the larger laser spot region. This is realized through the use of a k-means clustering algorithm to optimally produce the target signal, based on a raw collected waveform and a scaled crosstalk signal from modified simulation. Finally, the authors detail an appropriate case study on a 7nm tech node defective microprocessor with their new technique, ultimately resulting in a successful failure analysis by locating a short fault.

- + This work has the potential for high impact in the field of failure analysis, due to the authors addressing a worrisome problem with a realistic attempt at a solution
- + Overall, the methodology appears sound, is supported by detailed justifications, and is validated by a sufficient experimental study
- + Figures and their captions are helpful and crucial to understanding this work
- + There are interesting implications for the field of hardware security too, since a deterrent for FA-tool based hardware attackers was the fact that it has been difficult to extract confidential data from smaller tech nodes
- In terms of novel content, a sizeable portion of this paper seems to rely on the combinational logic analysis algorithms already addressed in the previous paper
- At first its confusing to completely understand the section describing the MC, it would be helpful to reiterate/emphasize that the activity map is dependent on time (the statement using the gate, source and drain regions of transistors as examples implies that the map highlights any region with a potential for activity, not actual activity in a particular moment)
- Minor writing errors (e.g. “pearson’s coefficient correlation”, “...normalized between -1 and 1, can then be plotted...”)

- A limitation to this technique could be its scalability and possibility of automation. Consider discussing these implications

Reviewer #2 (Remarks to the Author):

Overall a very good paper with very important content.

Issues:

- line 48 2nd paragraph intro:

The abbreviation "CCC" needs more explanation. How it is composed from the topics and words you mention. CCC is normally used for Collapsed cone convolution.

Example: Collapsed Cone Convolution and analytical anisotropic algorithm dose calculations compared to VMC++ Monte Carlo simulations in clinical cases, F Hasenbalg 1 , H Neuenschwander, R Mini, E J Born, Phys Med Biol, 2007 Jul 7;52(13):3679-91.

doi: 10.1088/0031-9155/52/13/002. Epub 2007 May 24.

- line 86, page 3 bottom: "from of" makes no sense as sentence

- Eq. (2) : $\sigma = 77\text{nm}$ eq. 2: from where? $\text{FWHM} / 0,42 = \sigma$ one more time divided by 2?

- line 208, page 8 bottom:

"Each cluster is formed by applying a k-means clustering algorithm¹⁶ on Eqn. 6, which is typically used to split a distribution into k number of clusters." –is not clear what “means” does to “k”. Is the motivation clear from beginning that you end up with the 2 clusters of high and low? If so, why do you write it in a more general way first?

Reviewer #1 (Remarks to the Author):

In this work, a new technique is presented to address the issue of decreasing tech node size alongside a relatively stagnant laser spot size/resolution in laser probing. Laser probing is important for locating and observing fabrication faults in ICs, so the technique has the potential to be impactful in the field of failure analysis. The technique, referred to by the authors as the “CCC algorithm”, consists of a three-part process in the form of Cross correlations, Clustering, and Combinational logic analysis. First, a combinational logic analysis waveform is generated, by constructing a spatial activity map over time from the target circuitry’s layout and incorporating equations pertaining to optical probing intensity. Then, the positional accuracy of the probing laser is finely-tuned to ensure that the user’s collected waveform genuinely corresponds to the transistors of interest. This is accomplished by using Pearson’s correlation coefficient to correlate collected waveforms at different positions with the expected simulation waveform. Third, a method is performed to extract a signal from a particular net within the larger laser spot region. This is realized through the use of a k-means clustering algorithm to optimally produce the target signal, based on a raw collected waveform and a scaled crosstalk signal from modified simulation. Finally, the authors detail an appropriate case study on a 7nm tech node defective microprocessor with their new technique, ultimately resulting in a successful failure analysis by locating a short fault.

Reviewer#1_+

- + This work has the potential for high impact in the field of failure analysis, due to the authors addressing a worrisome problem with a realistic attempt at a solution
- + Overall, the methodology appears sound, is supported by detailed justifications, and is validated by a sufficient experimental study
- + Figures and their captions are helpful and crucial to understanding this work
- + There are interesting implications for the field of hardware security too, since a deterrent for FA-tool based hardware attackers was the fact that it has been difficult to extract confidential data from smaller tech nodes

Author response:

We thank the reviewer for the positive feedback with regards to the technical soundness, the quality of the write up and underscoring the impact of this work to the industry.

Reviewer#1_Q1

- In terms of novel content, a sizeable portion of this paper seems to rely on the combinational logic analysis algorithms already addressed in the previous paper

Author response

It appears that we did not sufficiently highlight the novelty of the work with the previous articles with regards to the combinational logic analysis algorithms. As significant part of the prior work comes from our team, we can highlight the key novelty aspects in this work as below.

- a. There are 4 references (Ref. 11, 17, 28, 30) that are foundational to this work. In Ref 11, we introduced modulation capacity. Then, we took a simplistic input stimulus of a periodic signal in a cascading inverter circuitry, which does not require significant computation. There were no SPICE models, no input sequences, and multi-state waveforms.
Ref 17 is a description of laser probing signals from simple two input combinational cells and includes no computations. This paper is simply an observation that laser probing waveforms from combinational

cells could have multiple levels. We had recommended building a library of waveforms for combinational cells, which in retrospect is quite a challenge due to the growing number and complexity of these cells. This was not practical with technology scaling.

- b. Refs 28 and Ref 30 are similar papers that describe use of simulations to predict shape of waveforms from combinational logic devices. These are the closest to the actual work presented here. However, none of these references demonstrated probe placement accuracy with a few-nanometres precision capability that we demonstrated in this current manuscript. Secondly, we achieved signal extraction from small deep subwavelength features, down to individual transistor gates, through newly introduced crosstalk subtraction algorithms. These results are the first demonstrations of its kind as well. Being able to extract signals from emitters 18 times smaller than the probe spot highlight the significance of the results.

Finally, we report another breakthrough method in defect prediction by analysing the laser probe waveforms. From our experience and knowledge, the closest work is the cell-aware diagnostics using user defined fault models, which is a purely electrical and diagnostics capability (**Hapke, F. et al. (2012). "Cell-aware Production test results from a 32-nm notebook processor", Proceedings - International Test Conference. <https://doi.org/10.1109/TEST.2012.6401533>**). There has been no report of introducing defects in SPICE models to predict the shape of laser waveforms to implicate the defect location. We believe that our process flow can significantly improve success rates in exposing defects at 7 nm technology and below, as defects become notoriously invisible to conventional methods of Failure Analysis.

- c. To address this comment, we have revised the manuscript by adding sentences to the second paragraph in introduction (Page 2, lines 47-49)

"We had previously developed combinational logic analysis (CLA) algorithms^{17,28-30} to predict the shape of the ensembled signals. While CLA was useful in simple combinational cells in mature technology, the methods described were neither scalable to smaller technology nodes, nor automatable, posing fundamental limitations to widespread adaptation."

Reviewer#1_Q2

- At first its confusing to completely understand the section describing the MC, it would be helpful to reiterate/emphasize that the activity map is dependent on time (the statement using the gate, source and drain regions of transistors as examples implies that the map highlights any region with a potential for activity, not actual activity in a particular moment)

Author response

- a. We thank the reviewer for the excellent suggestion, and it has allowed us to improve the paper in the sections surrounding the explanation of MC. It is true that we had combined both the timing information (timing map) and the modulation propensity of different active elements in the physical layout (modulation map). This means, that the modulation map changes with input sequences / time. We hope that the significant changes made in sections titled "computation of modulation capacity" can better describe MC
- b. To address this comment, we have revised the manuscript on Page 5, lines 114-122 and added a sub figure Fig. 2 (a)

"The process can be explained with the help of the schematic shown in Fig. 2 (a). First, we construct a high resolution (1 nm in x and y) two-dimensional model of the circuitry's physical layout called the modulation map (MM) which represents the spatial distribution of the relative modulation the device would impart to the optic probe. The map includes all active areas where relative changes to free carriers are expected, such as the gate,

source and drain regions of each transistor, and is normalized to the PMOS drain as a convention. Second, we construct a timing map (TM) of the same circuitry, which combines the electrical models of the circuitry and test stimuli sequences. Since the test vectors change with time, the timing map can be construed as a three-dimensional matrix, with the third dimension representing time. A piecewise multiplication of the modulation map with the timing map is a time-varying three-dimensional matrix called the modulation capacity (MC)¹¹ and is described in Eqn. 2. The MC thus represents the influence of the physical device on the optic probe over time.

- c. Additionally, we have also edited equations 6 and added equation 2 to better explain MC and computation of the dynamic reflected power.

$$MC(x, y, t) = MM(x, y) \cdot TM(x, y, t) \quad (2)$$

$$Dynamic\ reflected\ power(x_0, y_0, t) \propto \iint_{x_i, y_j=0}^n MC(x_i, y_j, t) \cdot L(x_i, y_j) dx \cdot dy \quad (6)$$

Fig. 1: Accurate probe positioning at the target transistor (a) Schematic of the MC and CLA simulation. MM contains the relative modulation that various components of the device impart to the optic probe L. TM changes with each input stimulus and contributes to changes in the modulation over time. (b) physical layout of cell (670 nm wide) under observation. The tiny squares represent the gate-fin overlap and the longer horizontal bars represent the source or drain segments of the fin. The polygons with the same color fluctuate with the same electrical signal and polygons shaded with diagonal textures are power supplies and do not toggle. The yellow 'X's represent the raw waveform positions (x_r) from which each of 23 optically probed (raw) waveforms are collected within the cell. Raw waveforms **r1** and **r2** are collected from positions indicated by the violet and green circles with diameters equal to the FWHM of the probe. (c) LP waveforms traces comparing **r1** and **r2** with CLA simulations. The best correlated simulation positions (x_s) for **r1** ($x_s = 471$ nm) and **r2** ($x_s = 497$ nm) are colored in black. Overlaid are also simulations at positions $x_s - 10$ nm, -20 nm and -30 nm in red and $x_s + 10$ nm, $+20$ nm and $+30$ nm in blue. (d) Heat map of the positional correlation between the raw waveforms (x_r) on X-axis (32 nm increments) with simulations (x_s) performed along the same axis with a resolution of 1 nm on Y-axis. A strong positive correlation is represented in deep red, and a strong inverse correlation is in deep blue. As the positions of the optic probe is changed along the X-axis, the physical positions correlate well with the corresponding positions in the simulation. The trend line connects the best correlated x_s with x_r . (e) Graph compares the variation in PCC index with the simulation positions for four physical positions (**pcc1-pcc4**) marked by colored rings in (d). In dotted black is the Gaussian optic probe distribution, which is sharper than the PCC distributions of each of the correlations, indicating that the method does not violate the optical diffraction limit of the system.

Reviewer#1_Q3

- Minor writing errors (e.g. "pearson's coefficient correlation", "...normalized between -1 and 1, can then be plotted...")

Author response

We thank the reviewer for helping us improve the paper by catching these writing errors. We apologize for having them in the manuscript, and hope that our fixes have helped address these points.

To address this comment, we have revised the manuscript on Page 6, line 144 and the second typo on Page 5, line 131.

"A multitude of LP waveforms (**raw**) is collected from different positions (x_r) within the cell and is correlated using Pearson's correlation coefficient (**PCC**) to CLA simulation waveforms on different positions (x_s) within the cell"

"It is normalized between -1 and 1, and can then be plotted as a function of time."

Reviewer#1_Q4

- A limitation to this technique could be its scalability and possibility of automation. Consider discussing these implications

Author response

We thank the reviewer for bringing up a key aspect that is fundamental to the method and adaptation in the industry and could be explored more as a discussion in the manuscript. We have since amended this section. It is quite rational to be concerned about the limitation of scalability of this technique, as there are a number of considerations. To begin with, the simulation load increases proportionally to the number of transistors in consideration. As technology scales, the numbers of transistors in consideration increases, and with it, the simulation time will be impacted. However, it is to be noted that the simulation time is quite small (usually a couple seconds per waveform on modern workstations). A second consideration for automation is in precursor files needed for the simulation, in terms of the cells in consideration, their layout and SPICE models, as well as the pattern information (input combinations). This is an area of ongoing development and collaboration. A third consideration is in automation of signal collection by the microscope, and subsequent correlation with the simulations. We have made several improvements in the hardware to speed up signal collection, and recursive automatic laser deflection to control the laser and completely automate the data generation. Due to the sensitivity of the nature of ongoing development with commercial implications, we are unable to completely elaborate the benefits to automation.

We have added to introduction Page 2, lines 55 and 56. We have also added a section limitation in Page 12 lines 299-304.

“The CCC allows scalability and can be combined with scripts to control the laser probe to automate, and extract volumes of useful information from the target circuitry.”

“Limitation

As technology is scaled, the simulation load increases proportionally with the number of transistors in consideration. As the run-time for the simulation of 7 nm cells take only about 1-2 seconds per waveform, the process is extremely scalable with technology. Iterative loops of probe placement deflections and cross-correlations can be automated to be used for large volume signal extractions effectively. However, to successfully apply these methods to a large area, the precursor files such as the physical layout, SPICE models and test pattern stimuli need to be prepared in advance and is an area of active development.”

Reviewer #2 (Remarks to the Author):

Overall a very good paper with very important content.

Author response:

We thank the reviewer for the positive feedback underscoring the impact of this work

Reviewer#2_Q1

- line 48 2nd paragraph intro:

The abbreviation "CCC" needs more explanation. How it is composed from the topics and words you mention. CCC is normally used for Collapsed cone convolution.

Example: Collapsed Cone Convolution and analytical anisotropic algorithm dose calculations compared to VMC++ Monte Carlo simulations in clinical cases, F Hasenbalg 1 , H Neuenschwander, R Mini, E J Born, Phys Med Biol, 2007 Jul 7;52(13):3679-91.

doi: 10.1088/0031-9155/52/13/002. Epub 2007 May 24.

Author response:

We thank the reviewer for bringing up the possibility of the acronym CCC being misconstrued. CCC stands for CLA, cross-correlations and clustering. We have coined the acronym with the first letters from the three major algorithms that we use in this work.

To address this comment, we have removed the usage of the acronym CCC in the abstract (Page1, line 19), and edited the introduction Page 2, line 49-50)

“Through algorithms introduced in this patented^{13,14} work, consisting of cross-correlations¹⁵, clustering¹⁶, and our previously developed combinational logic analysis¹⁷, we achieved beam positioning accuracy to better than 10 nm, extracted electrooptic waveforms from a node of a group of transistors”

“In this work^{14,29}, we introduce an algorithmic method consisting of CLA, cross-correlations¹⁵, and clustering¹⁶ named CCC (one C for each of the three aforementioned algorithms) that leverages on the known available electrical information to aid in aligning the optic probe and extracting the signals of interest.”

Reviewer#2_Q2

- line 86, page 3 bottom: "from of" makes no sense as sentence

Author response:

We thank the reviewer for helping us improve the paper by catching these writing errors. We apologize for having them in the manuscript, and hope that our fix (removed the word of) have helped address these points.

“While the electrical state of a digital circuitry is binary, i.e., either ON or OFF, these intermediate intensity levels occur due to the superposition of signals from multiple transistors (or electrooptical crosstalk) within the optic probe, and their respective modulation capacity (MC) as discussed later.”

Reviewer#2_Q3

- Eq. (2) : sigma =77nm eq. 2: from where? FWHM / 0,42 = sigma one more time divided by 2?

Author response:

We thank the reviewer for pointing out that there is a lack of clarity in representation of a 2D system which affects the derivation of Eq. 2. If we consider the two dimensions, we have to treat σ in x and y axes separately. We have therefore simplified description to a 1D which is valid for circularly polarized systems that are used in this work. We derive σ through comparison of the full width half max to a typical gaussian distribution as follows.

$$FWHM = 0.5 \frac{\lambda}{NA} \quad -(1)$$

$$L = \frac{1}{\sqrt{2\pi\sigma^2}} e^{-\frac{((x-x_0)^2+(y-y_0)^2)}{2\sigma^2}} \quad -(2)$$

We can simplify this equation to one dimension by observing the point spread function along just the x axis ($y = y_0$), so the equation simplifies as below.

$$L = \frac{1}{\sqrt{2\pi\sigma^2}} e^{-\frac{((x-x_0)^2)}{2\sigma^2}} \text{ (for one axis)} \quad -(3)$$

, and for a normalized when $L = \frac{1}{2}$ which is the FWHM condition, this equation will reduce to

$$\frac{1}{2} = \frac{1}{\sqrt{2\pi\sigma^2}} e^{-\frac{((x-x_0)^2)}{\sigma^2}} \quad -(4)$$

When we take natural log, rearrange the terms to relate the $x-x_0$ to σ

$$x - x_0 = \sigma\sqrt{2\ln 2} \quad -(5)$$

Since the FWHM is twice of $x - x_0$,

$$\text{FWHM} = \sigma \cdot 2\sqrt{2\ln 2} \quad (6)$$

From Equations 1 and 6,

$\sigma = 0.21 \frac{\lambda}{NA}$ which is approximately 77nm for wavelength (1064nm) and NA (2.9).

To address this question in the manuscript, we have made changes to the description in Page 5, lines 123-127, Eqn. 3 and Eqn. 5

“Next, we model the optic probe as a Gaussian distribution. The full width at half-maximum spread (FWHM) of the probe is related to the wavelength (λ) and numerical aperture (NA) using Abbe’s criteria, as shown in Eqn. 3²². Moreover, the standard deviation (σ), is related to the FWHM described in the same equation³⁶. Therefore, σ can be represented in terms of λ and NA as shown in Eqn. 4. For the system used in this work, $\lambda = 1064$ nm, $NA = 2.9$, FWHM is 183 nm, and $\sigma = 77$ nm. The optical intensity distribution (L) centered at position (x_0, y_0) is shown in Eqn. 5.

$$\text{FWHM} = 0.5 \frac{\lambda}{NA} = \sigma \cdot 2\sqrt{2\ln 2} \quad (3)$$

$$L = \frac{1}{\sqrt{2\pi\sigma^2}} e^{-\frac{(x-x_0)^2+(y-y_0)^2}{2\sigma^2}} \quad (5)$$

Reviewer#2_Q4

- line 208, page 8 bottom:

"Each cluster is formed by applying a k-means clustering algorithm¹⁶ on Eqn. 6, which is typically used to split a distribution into k number of clusters." –is not clear what “means” does to “k”. Is the motivation clear from beginning that you end up with the 2 clusters of high and low? If so, why do you write it in a more general way first?

Author response:

We thank the reviewer in pointing out that our usage of k-means clustering algorithm in our application is not explained adequately. In the examples we have used, based on our SPICE models, the nets take only two values, 0 and 1. K-means is a generic AI algorithm as you have also pointed out, is commonly used to bifurcate a distribution into k clusters. In our example we only required two levels. We determined that the k-means clustering had the versatility required for use in our algorithm. It is possible to use a brute force method without AI, by manually applying scaling parameters too, but the method could not be automated. We have modified the text in this section to improve the explanation further, and hope it will address the reviewers concerns.

To address this question in the manuscript, we have made changes to the description in Page 9, lines 215-223, Eqn. 3 and Eqn. 5

“As the nets of interest in our circuitry is expected to have only two levels under normal operation, the C_{net} can be determined through a combination of clustering algorithms¹⁶ that separate the waveform into two clusters while minimizing a loss function ($Loss_1$) shown in Eqn. 9. Each cluster is formed by applying a k-means clustering algorithm¹⁶ to R_{net} , which is commonly used to split a distribution into k number of clusters.”

“Optimizing recursively between Eqns. 8 and 9 using gradient descent³⁸, this artificial intelligence (AI) algorithm can identify the ideal C_{net} for the smallest $Loss_1$. Alternate methods such as linear search with coarse and fine step sizes or even trial and error may also be used to identify a suitable C_{net} .”

REVIEWERS' COMMENTS

Reviewer #1 (Remarks to the Author):

In this work, a new technique is presented to address the issue of decreasing tech node size alongside a relatively stagnant laser spot size/resolution in laser probing. Laser probing is important for locating and observing fabrication faults in ICs, so the technique has the potential to be impactful in the field of failure analysis. The technique, referred to by the authors as the “CCC algorithm”, consists of a three-part process in the form of Cross correlations, Clustering, and Combinational logic analysis. First, a combinational logic analysis waveform is generated, by constructing a spatial activity map over time from the target circuitry’s layout and incorporating equations pertaining to optical probing intensity. Then, the positional accuracy of the probing laser is finely-tuned to ensure that the user’s collected waveform genuinely corresponds to the transistors of interest. This is accomplished by using Pearson’s correlation coefficient to correlate collected waveforms at different positions with the expected simulation waveform. Third, a method is performed to extract a signal from a particular net within the larger laser spot region. This is realized through the use of a k-means clustering algorithm to optimally produce the target signal, based on a raw collected waveform and a scaled crosstalk signal from modified simulation. Finally, the authors detail an appropriate case study on a 7nm tech node defective microprocessor with their new technique, ultimately resulting in a successful failure analysis by locating a short fault.

- + This work has the potential for high impact in the field of failure analysis, due to the authors addressing a worrisome problem with a realistic attempt at a solution
- + Overall, the methodology appears sound, is supported by detailed justifications, and is validated by a sufficient experimental study
- + Figures and their captions are helpful and crucial to understanding this work
- + There are interesting implications for the field of hardware security too, since a deterrent for FA-tool based hardware attackers was the fact that it has been difficult to extract confidential data from smaller tech nodes
- + Reliance on previous work in CLA algorithms has been addressed
- + The new revision has greatly improved the clarity of the MC description and time-reliance, including a helpful graphic, additional discussion, and modification to the previous equations
- + Minor writing errors have been addressed
- + Scalability and automation factors are discussed

Reviewer #2 (Remarks to the Author):

Thank you very much for your detailed work with the reviewer's comments. The article is in excellent shape.

REVIEWERS' COMMENTS

Reviewer #1 (Remarks to the Author):

In this work, a new technique is presented to address the issue of decreasing tech node size alongside a relatively stagnant laser spot size/resolution in laser probing. Laser probing is important for locating and observing fabrication faults in ICs, so the technique has the potential to be impactful in the field of failure analysis. The technique, referred to by the authors as the “CCC algorithm”, consists of a three-part process in the form of Cross correlations, Clustering, and Combinational logic analysis. First, a combinational logic analysis waveform is generated, by constructing a spatial activity map over time from the target circuitry’s layout and incorporating equations pertaining to optical probing intensity. Then, the positional accuracy of the probing laser is finely-tuned to ensure that the user’s collected waveform genuinely corresponds to the transistors of interest. This is accomplished by using Pearson’s correlation coefficient to correlate collected waveforms at different positions with the expected simulation waveform. Third, a method is performed to extract a signal from a particular net within the larger laser spot region. This is realized through the use of a k-means clustering algorithm to optimally produce the target signal, based on a raw collected waveform and a scaled crosstalk signal from modified simulation. Finally, the authors detail an appropriate case study on a 7nm tech node defective microprocessor with their new technique, ultimately resulting in a successful failure analysis by locating a short fault.

- + This work has the potential for high impact in the field of failure analysis, due to the authors addressing a worrisome problem with a realistic attempt at a solution
- + Overall, the methodology appears sound, is supported by detailed justifications, and is validated by a sufficient experimental study
- + Figures and their captions are helpful and crucial to understanding this work
- + There are interesting implications for the field of hardware security too, since a deterrent for FA-tool based hardware attackers was the fact that it has been difficult to extract confidential data from smaller tech nodes
- + Reliance on previous work in CLA algorithms has been addressed
- + The new revision has greatly improved the clarity of the MC description and time-reliance, including a helpful graphic, additional discussion, and modification to the previous equations
- + Minor writing errors have been addressed
- + Scalability and automation factors are discussed

Reviewer #2 (Remarks to the Author):

Thank you very much for your detailed work with the reviewer's comments. The article is in excellent shape.